# Cytokine-Induced JAK2-STAT3 Activates Tissue Regeneration under Systemic or Local Inflammation

**DOI:** 10.3390/ijms23042262

**Published:** 2022-02-18

**Authors:** Young Kyu Kim, Ju Young Lee, Han Na Suh

**Affiliations:** Animal Model Research Group, Korea Institute of Toxicology, 30 Baekhak1-gil, Jeongeup 56212, Jellabuk-do, Korea; youngkyu.kim@kitox.re.kr (Y.K.K.); juyoung.lee@kitox.re.kr (J.Y.L.)

**Keywords:** LPS, ovalbumin, NF-κB, cytokines, JAK2/STAT3, tissue regeneration

## Abstract

We investigated the immune response mechanisms under systemic and local inflammation using mouse models whereby lipopolysaccharide (LPS) was administered intraperitoneally to induce systemic inflammation, and epicutaneous sensitization with ovalbumin was used to induce local inflammation. LPS increased the immune cell infiltration in the cardiac muscle near the aorta, alveoli, hepatic sinusoid, renal interstitium, and the submucosal layer of the duodenum. Similarly, ovalbumin increased the abundance of macrophages in the skin. Both LPS and ovalbumin induced NF-κB p65 and IκBα phosphorylation, as well as the expression of NF-κB target genes (*TLR4*, *IL6*, and *TNF**α*). Additionally, both LPS and ovalbumin led to an increase in the absolute IL-1β, IL-6, and TNFα serum levels and cytokine-related janus kinase 2 (JAK2)/signal transducer and activator of transcription 3 (STAT3) phosphorylation. Moreover, the activated JAK2/STAT3 signaling increased the number of Ki67-positive cells (proliferating cells) and development pathway target gene expression (regeneration) in the inflammation models. In conclusion, LPS and ovalbumin increase immune cell infiltration in tissues, NF-κB activation, cytokine levels in serum, cytokine-stimulated JAK2/STAT3 signaling, and tissue regeneration.

## 1. Introduction

Inflammation is our body’s response to protect itself against invading pathogens or tissue injury [1]. Acute inflammation is an immediate immune response that triggers cytokines and chemokines to promote the migration of neutrophils to the injured or infected area to eliminate pathogens and regenerate tissues [2]. When the acute response fails to remove the pathogen, chronic inflammation sets in and can lead to metabolic diseases such as coronary heart disease, type 2 diabetes, and rheumatoid arthritis caused by aberrant proinflammatory cytokines [3]. Common methods for diagnosing inflammation include hematology, serum chemistry, histopathology, or assessing patient symptoms. However, these analytic tools are unable to detect the inflammation-induced dynamic response of immunity.

Previous studies have established inflammatory rodent models by administering specific antigens. To study systemic inflammation, exogenous administration of a toxin such as lipopolysaccharide (LPS) or a viable pathogen (inoculation of live bacteria such as *Escherichia coli*) is applied to the mouse models [4]. LPS stimulates host immune cells (monocytes, macrophages, and B cells) through toll-like receptor 4 (TLR4) and enhances interleukin (IL)-1β, IL-6, and tumor necrosis factor α (TNFα) expression [5]. Ovalbumin epicutaneous sensitization is widely used to induce atopic dermatitis, skin disorders, and the antigen-specific immune response [6,7]. Ovalbumin initiates the increase of IgE, followed by myeloid immune cell infiltration and cytokine secretion. Atopic dermatitis is a chronic inflammatory skin disease that evolves into systemic inflammation [8,9], which suggests the existence of a common signaling pathway between local and systemic inflammation. However, the link between local and systemic inflammation remains unknown.

The janus kinase (JAK)-signal transducer and activator of transcription (STAT) pathway mediates the immune response, hematopoiesis, and cytokine and growth factor synthesis. Once a cytokine binds to its own membrane receptor, the intracellular domain-associated JAK is activated via transphosphorylation [10]. JAK phosphorylation changes the docking site of STAT at the tail region of the receptor and mediates STAT phosphorylation and nuclear translocation. The translocated STAT then binds to various promoters to regulate gene expression. JAK/STAT signaling is essential for tissue regeneration after damage [11]. However, it is not clear whether the JAK/STAT pathway is involved in regeneration after LPS-induced acute or chronic systemic inflammation and ovalbumin-induced local inflammation. In this study, we determine the correlation between systemic and local inflammation and the role of the JAK/STAT signaling pathway in tissue regeneration.

## 2. Results

### 2.1. Validation of Systemic and Local Inflammation Mouse Models Using Histopathology

To determine the mechanism of systemic and local inflammation, we developed mouse models for systemic or local inflammation (Figure 1A). The inflammatory models were validated by histopathology. LPS induced interstitial inflammation, immune cell infiltration, and congestion in pulmonary alveoli but did not induce any phenotypic changes in the left ventricle cardiac muscle or the crypt–villus structures of the duodenum. Immune cell infiltration was also found in the hepatic tissue near the central vein in both the chronic LPS and acute LPS mice. Additionally, LPS induced renal proximal tubule injury and lymphoid infiltration in the kidney cortex (Figure 1B). Ovalbumin-sensitized skin showed epidermal hyperplasia and hyperkeratosis (Figure 1C). These results confirmed that both the LPS-induced systemic and the ovalbumin-induced local inflammation mouse models can be used to investigate the mechanisms of systemic and local inflammation.

### 2.2. Detection of Immune Cells through Immunohistochemistry to Confirm Inflammation

Macrophages are immune cells that reside in various tissues. To determine the abundance and distribution of immune cells in tissues, we performed immunohistochemistry using an antibody against F4/80, a mouse macrophage marker [12]. LPS increased the abundance of macrophages in cardiac muscles near the aorta, alveoli, hepatic sinusoid, renal interstitium, and submucosal layer of the duodenum. Moreover, the tissues in LPS-induced acute inflammation had a greater abundance of macrophages than those of chronic inflammation (Figure 2A). Ovalbumin also enhanced macrophage infiltration at the skin dermis (Figure 2B). These results showed that increases in immune cell infiltration can occur even in the absence of histopathological phenotypes. This suggests that immune cell detection in tissues is a more sensitive method for determining inflammation.

### 2.3. Increased NF-κB Activation and NF-κB Target Gene Expression under Inflammation

As NF-κB is a transcription factor essential for the inflammatory response, NF-κB activation was detected by immunoblot. Both acute and chronic LPS increased NF-κB p65 serine 236 phosphorylation (NF-kB p65 nuclear translocation) and IκBα serine 36 phosphorylation (I-kBα degradation) in the heart, lungs, liver, and skin. However, only acute LPS increased NF-κB p65 and IκBα phosphorylation in the kidneys and duodenum. Of note, ovalbumin epicutaneous sensitization enhanced the activation of NF-κB pathways in the liver and skin (Figure 3A). To determine the effect of NF-κB activation on the downstream pathways in both systemic and local inflammation, NF-κB target genes were analyzed by qRT-PCR. The expression of *TLR4*, *IL6*, and *TNF**α* were upregulated under systemic or local inflammation in the heart, liver, and skin. *TLR4* and *TNF**α* were enhanced in the lungs under systemic and local inflammation. *TNF**α* was increased in the duodenum under systemic and local inflammation, while *TNF**α* was increased in the kidneys only under chronic systemic inflammation (Figure 3B). These results indicate that both LPS-induced systemic inflammation and ovalbumin-induced local inflammation upregulate the expression of NF-κB target genes in the heart, lungs, liver, duodenum, and skin, suggesting a strong correlation between local and systemic inflammation. Additionally, each organ shows the upregulation of specific target genes that is consistent for both LPS-induced and ovalbumin-induced inflammation (heart, liver, and skin: *TLR4*, *IL6*, and *TNF**α*; lungs: *TLR4* and *TNF**α*; and kidneys and duodenum: *TNF**α*).

### 2.4. JAK2- STAT3-Induced Tissue Regeneration after Inflammatory Damage

Inflammation plays an important role in immune activation and tissue regeneration. To determine the effect of systemic and local inflammation on the cytokine level, the abundance of proinflammatory cytokines was measured by ELISA. Chronic and acute LPS-induced inflammation and ovalbumin epicutaneous sensitization increased the serum levels of IL-1β and IL-6, whereas only acute LPS exposure increased the TNFα levels. Following acute LPS exposure, the IL-6 levels were over 800 times higher than those in the controls (Figure 4A–C). To determine whether LPS-induced systemic and ovalbumin-induced local inflammations are recovered through tissue regeneration, cytokine-related JAK/STAT signaling was assessed. JAK2 and STAT3 phosphorylation increased following acute LPS exposure in the heart, lungs, liver, kidneys, duodenum, and skin (Figure 4D), which possibly indicates tissue regeneration. Additionally, acute LPS exposure increased the number of Ki67-positive cells in tissues, which points to the proliferative capacity of the organs (Figure 4E). Ovalbumin sensitization led to a partial increase in the phosphorylation levels of JAK2 and STAT3 in the skin compared to acute LPS exposure, which suggests the involvement of other regenerative pathways in the skin. Additionally, ovalbumin-treated skin showed an increase in the Ki67-positive cell population in both the epidermis and dermis (Figure 4F). In the analysis of five major development pathway target gene expressions, sonic hedgehog target gene *Gli1* was mostly upregulated in the heart. Wnt/β-catenin target gene *CD44* was highly upregulated in the lungs and liver. Notch target gene *Hes1* was strongly upregulated in the kidneys. Hippo target gene *Ctgf* was strongly upregulated in the duodenum. BMP target gene *Ptprg* was upregulated in the skin (Figure 4G). Those gene upregulations provided strong evidence of tissue regeneration after inflammatory damage. Overall, we found that LPS and ovalbumin increase immune cell infiltration in tissues, NF-κB activation, cytokine levels in the serum, cytokine-stimulated JAK2/STAT3 signaling activation, and tissue regeneration (Graphical Abstract).

## 3. Discussion

In this study, we demonstrated that cytokine-induced JAK/STAT signaling activation mediates tissue regeneration after systemic or local inflammation. As rodents are an attractive animal model due to the associated low cost and ease of handling [13], we developed inflammatory mouse models to investigate the correlation between systemic and local inflammation, as well as the tissue regeneration mechanism. We evaluated inflammation by histopathology and direct immune cell detection in tissues. As seen in Figure 1B, no phenotypic abnormalities were detected in the heart or duodenum under LPS induction. Each organ has a different parenchymal cell turnover rate. For example, 3 to 4% of cardiomyocytes are renewed every year [14], and intestinal epithelial cells are replaced from the crypt to villi tip every 4 to 5 days [15]. These differences might affect the sensitivity and resistance to damage, as well as phenotypical changes. Interestingly, macrophage infiltration was found in the heart and duodenum (Figure 2A), indicating an activated immune response. Recently, we have reported that immune cell detection in a tissue is a more precise method of determining inflammation [16]. Once inflammation occurs, macrophages are recruited to the injured or infected area and release cytokines. Then, parenchymal cells in the target organ decide the cell fates. Thus, the presence of immune cells can be used for inflammation detection in the early stage.

LPS binds to TLR4, which then transduces NF-κB signaling. In this study, we identified an increase in NF-κB activation and NF-κB target gene expression under chronic and acute LPS treatment (Figure 3). Specifically, the expression of *TLR4, IL6,* and *TNF**α* increased in the heart, liver, and skin; *TLR4,* and *TNF**α* increased in the lungs; and *TNF**α* was enhanced in the kidneys and duodenum after LPS treatment. These results suggest that LPS fine-tunes specific NF-κB target gene expressions in each organ. Ovalbumin skin sensitization increased NF-κB activation and the expression of NF-κB target genes, which suggests a correlation of ovalbumin and the NF-κB signaling pathways. This is in line with previous studies that reported that ovalbumin activates the TLR4 pathway and NF-κB in airway inflammation [17,18,19]. Moreover, the inhibition of NF-κB attenuates ovalbumin-induced allergic asthma [20]. Furthermore, we found that the epicutaneous application of ovalbumin leads to the upregulation of NF-κB target genes not only in the skin but also in the heart, lungs, liver, and duodenum. LPS also led to the upregulation of NF-κB target genes in the skin. These results suggest that systemic and local inflammation have reciprocal actions. Previous studies supported our findings that epicutaneous ovalbumin sensitization-stimulated airway inflammation is reduced by *Il22* gene deletion or IL-22-neutralizing antibody pretreatment [21].

Inflammation is an essential player in tissue regeneration through the removal of dead cells and secretion of biomolecules. As shown in Figure 3B, we identified an increase in cytokine mRNA under inflammation. Along with cytokine mRNA upregulation, increase of the absolute cytokine levels in serum was also detected (Figure 4A–C). Those cytokines, IL-6 and TNFα, are crucial for tissue homeostasis maintenance through the JAK/STAT signaling pathway [22,23]. Furthermore, the cytokine/JAK/STAT pathway is required for differentiation or regeneration of the midgut [24]. In addition, STAT3 induces intestinal [11] or hepatic regeneration by regulating cell cycle progression [25]. Thus, cytokine/JAK/STAT signal transduction appears to be an important regenerative pathway following inflammatory damage. Acute LPS exposure increases JAK2 and STAT3 phosphorylation, which suggests that potential tissue regeneration follows. Ovalbumin-treated skin showed only a partial increase in JAK2/STAT3 phosphorylation, which suggests the existence of another regenerative pathway. Usually, tissue regeneration is mediated by adult stem cells [26,27,28,29]. To display tissue regeneration, it is the best to find self-renewing stem cells in the damaged tissue. However, as the adult stem cell marker is still controversial (Lgr5, Bmi1, TERT, Axin2, etc.), adult stem cell detection has limitations. In this study, we detected Ki67+ proliferating cells. Some of the Ki67+ cells might be adult stem cells, and the rest of the Ki67+ cells might be parenchymal cells or others. We compared the number of Ki67+ cells in the control vs. treated groups and found an increased number, which showed their proliferating status clearly. Additionally, there was an increase of major development pathway target gene expression, which was strongly connected with tissue regeneration [29] (Figure 4G). In this study, we identified LPS-induced systemic and ovalbumin-induced local inflammation as having a direct influence on the target organs through TLR4 and NF-κB-related signaling pathways. NF-κB-induced cytokines regulate JAK2/STAT3 signaling and tissue regeneration. This study uncovered the common inflammation-mediated signaling pathway and its related tissue repair signaling pathway. In conclusion, LPS or ovalbumin increases the immune cell infiltration in tissues, NF-κB activation, cytokines in the serum, cytokine-stimulated JAK2/STAT3 signaling activation, and tissue regeneration.

## 4. Materials and Methods

### 4.1. Experimental Animals

Fifty C57BL/6 mice (*Mus muscularis*; eight weeks old weighing 20–25 g; OrientBio, Seongnam, Korea) were used. Six mice were randomly selected based on body weight for each experimental group. Lipopolysaccharide (LPS; from *Escherichia coli* O55:B5; Sigma, Munich, Germany) was administered intraperitoneally to induce systemic inflammation. The chronic systemic inflammation group was injected seven times with 0.25-mg/kg LPS, and the acute systemic inflammation group was injected with 2-mg/kg LPS once. Ovalbumin (Ova; from chicken egg white; Sigma) treatment was used to induce local atopic dermatitis-like skin inflammation. Specifically, 100-μL ovalbumin (1 mg/mL) was placed on the shaved back skin of mice using a gauze patch (1 × 1 cm^2^), which was secured with Tegaderm (3M Health Care, St Paul, MN, USA) and changed daily for seven days. Mice were housed under a 12:12-h light–dark cycle with lights on at 8 a.m. Water and food were provided ad libitum. All the animal experiments were conducted under the Institutional Animal Care and Use Committee guidelines of the Korea Institute of Toxicology (IACUC approval # 19-1-0192).

### 4.2. Histological Analysis

#### Histopathology

Tissues (heart, lungs, liver, kidneys, duodenum, and skin) were fixed in 10% neutral buffered formalin (NBF) overnight and embedded in paraffin. Tissue samples were sectioned (5 μm), deparaffinized, and stained with hematoxylin and eosin. To identify structural abnormalities, the left ventricle of the heart, pulmonary alveoli, renal cortex (glomeruli and proximal tubules), hepatic tissues near the central vein, and epithelial structures of the duodenum were examined. Abnormal regions are indicated in the figures with a black arrowhead.

### 4.3. Immunohistochemistry

Tissues were fixed in 10% NBF overnight and embedded in paraffin. Tissue samples were sectioned (5 μm), deparaffinized, processed for antigen retrieval, blocked, and then incubated with the target primary antibody, followed by either a peroxidase-conjugated or a fluorescence-conjugated secondary antibody. Samples were mounted and photographed using either microscopy (Leica DM2700) or confocal microscopy (ZEISS LSM800). For samples with the peroxidase-conjugated secondary antibody, 3, 3′-diaminobenzidine (Vector Laboratories, Burlingame, CA, USA) substrate was added, followed by hematoxylin for nuclear counterstaining. For samples with the fluorescence-conjugated secondary antibody, 4′, 6-diamidino-2-phenylindole was used for nuclear counterstaining. All images were captured using the same exposure time.

### 4.4. Immunoblot

Tissues were dissolved in radioimmunoprecipitation assay buffer containing protease/phosphatase inhibitor. Proteins in cell lysates (20–40 μg) were resolved by 6–15% gradient SDS-polyacrylamide gel electrophoresis and transferred to polyvinylidene fluoride membranes. Each membrane was incubated with 1% bovine serum albumin and probed with the primary and HRP-conjugated secondary antibodies listed in Table 1. The bands were visualized by enhanced chemiluminescence, and images were acquired for the quantitative analysis with ChemiDoc™ XRS^+^ System (Bio-Rad, Hercules, CA, USA).

### 4.5. Gene Analysis

For the gene expression analysis, total RNA was isolated using the RNeasy Mini Kit (Qiagen) and subjected to reverse transcription using iScript RT Supermix for quantitative real-time polymerase chain reactions (qRT-PCR; Bio-Rad). qRT-PCR was performed using intron-spanning primers. 18s was used as an endogenous control for normalization, and the expression levels were calculated using the ΔΔCT method. In Figure 3B and Figure 4G, values are displayed using heatmap software (bar.utoronto.ca/, accessed on 2 August 2019). The primer sequences are listed in Table 2.

### 4.6. Enzyme-Linked Immunosorbent Assay (ELISA)

To measure the absolute cytokine levels, serum was obtained from mice at day 8 of the experimental treatment. Mouse IL-1β (MLB00C; R&D), IL-6 (M6000B; R&D), and TNFα (MTA00B; R&D) were measured by the ELISA method according to the manufacturer’s protocol.

### 4.7. Statistical Analyses

Statistical analysis was performed using GraphPad Prism 7 software, and the differences between control and treated group were analyzed using a Student’s *t*-test. Values were indicated as the mean ± standard deviation (SD). *p*-values < 0.05 were considered significant. Every experiment included six biological replicates and three experimental replicates unless otherwise mentioned in the figure legends.

## Figures and Tables

**Figure 1 ijms-23-02262-f001:**
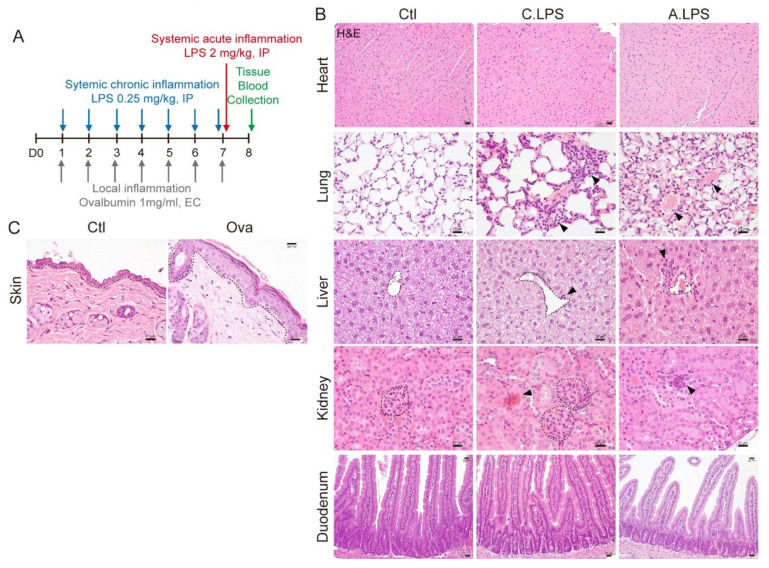
Histopathological analysis. (**A**) Experimental schematic. For chronic systemic inflammation, 0.25-mg/kg lipopolysaccharide (LPS) was administered intraperitoneally (IP) seven times daily. For acute systemic inflammation, 2-mg/kg LPS was administered IP once at day 7. For local skin inflammation, the skin was sensitized with 1-mg/mL ovalbumin (Ova) epicutaneously seven times daily. Blood and tissue were collected for further study. (**B**) Hematoxylin and eosin (H&E) staining. Central vein in the liver and glomeruli in the kidneys are indicated with a black dotted line. Abnormal lesions are indicated with a black arrowhead. Scale bar = 20 μm. (**C**) H&E staining. Epidermis is indicated with a black dotted line. Scale bar = 20 μm. Representative images are shown; *n* ≥ 3.

**Figure 2 ijms-23-02262-f002:**
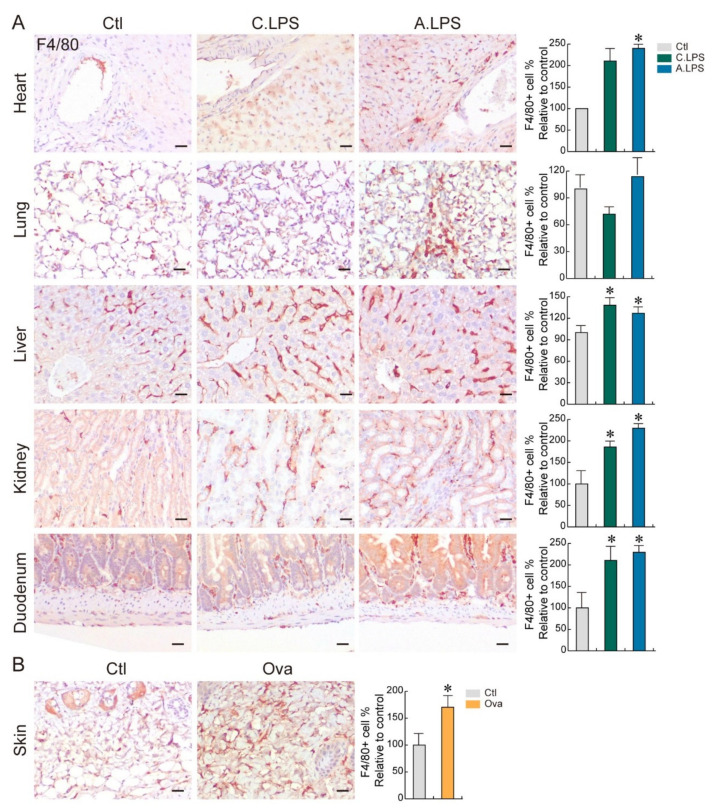
Immune cell distribution. (**A**,**B**) Immunohistochemistry. Population and distribution of F4/80 (macrophages) were examined in the heart, lungs, liver, kidneys, duodenum (**A**), and skin (**B**). Representative images are shown; *n* ≥ 3. Quantification of the immune cell number is shown as the mean ± SD; * *p* < 0.05 relative to the control. Scale bar = 20 μm.

**Figure 3 ijms-23-02262-f003:**
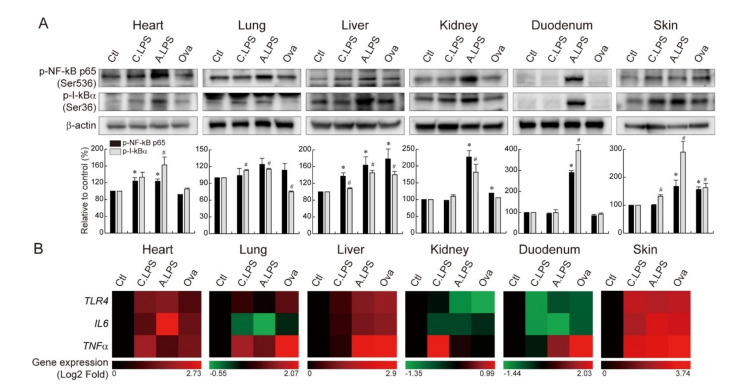
NF-κB activation and NF-κB target gene upregulation. (**A**) Immunoblot. NF-κB p65 serine 236 phosphorylation and IκBα serine 36 phosphorylation were detected in the heart, lungs, liver, kidneys, duodenum, and skin of animals with chronic or acute systemic inflammation and local inflammation. Bands were normalized by β-actin. Quantification of the bands is shown as relative to the control (mean ± SD); * *p* < 0.05 relative to the p-NF-κB p65 control; ^#^
*p* < 0.05 relative to the p-IκBα control. (**B**) qRT-PCR and heatmap. *TLR4*, *IL6*, and *TNF**α* mRNA expressions were analyzed in the heart, lungs, liver, kidneys, duodenum, and skin of animals with chronic or acute systemic and local inflammation. The red color indicates gene upregulation, and the green color indicates gene downregulation.

**Figure 4 ijms-23-02262-f004:**
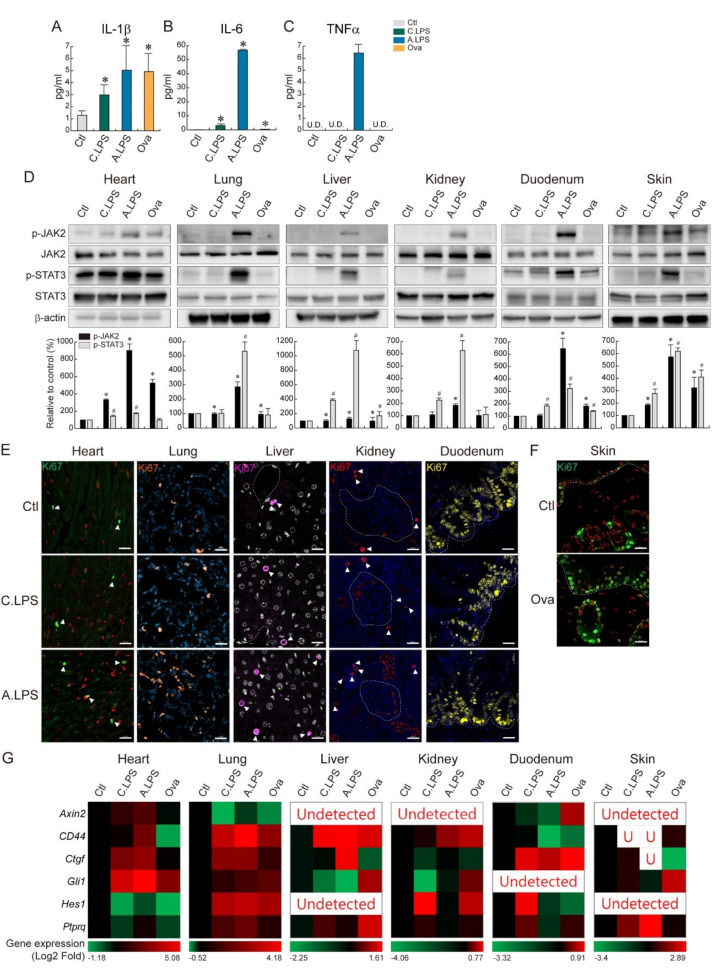
Tissue regeneration through cytokine-related JAK2-STAT3 activation. (**A**–**C**) ELISA. Absolute cytokine levels (pg/mL) were measured by ELISA. (**A**) IL-1β, (**B**) IL-6, and (**C**) TNFα. U.D. = undetected. The graph represents the absolute value of cytokine with the mean ± SD; * *p* < 0.05 relative to the control. (**D**) Immunoblot. JAK2 and STAT3 phosphorylation were detected in the heart, lungs, liver, kidneys, duodenum, and skin. Bands were normalized by total JAK2 or total STAT3. Quantification of the bands are shown as relative to the control (mean ± SD); * *p* < 0.05 relative to the p-JAK2 control; ^#^
*p* < 0.05 relative to the p-STAT3 control. (**E**,**F**) Immunohistochemistry. Population and distribution of Ki67 (proliferating cells) were examined in the heart, lungs, liver, kidneys, duodenum (**E**), and skin (**F**). Ki67-positive cells were marked with a white arrowhead. Representative images are shown; *n* ≥ 3. Scale bar = 20 μm. (**G**) qRT-PCR and heatmap. *Axin2, CD44, Ctgf, Gli1, Hes,* and *Ptprq* mRNA expressions were analyzed in the heart, lungs, liver, kidneys, duodenum, and skin of animals with chronic or acute systemic and local inflammation. The red color indicates gene upregulation, and the green color indicates gene downregulation.

**Table 1 ijms-23-02262-t001:** Antibodies used for the immunoblot.

Antibody	Company & Cat#	Dilution
p-NF-KB p65 (Ser536)	Cell Signaling #3033	1:1000
p-I-KBα (Ser36)	Abcam ab133462	1:1000
p-JAK2	Cell Signaling #3776	1:1000
JAK2	Cell Signaling #3230	1:1000
p-STAT3	Cell Signaling #9145	1:2000
STAT3	Cell Signaling #4904	1:2000
β-actin	Santa Cruz sc-47778	1:5000

**Table 2 ijms-23-02262-t002:** Primers used for the quantitative real-time PCR.

Gene Symbol	Primer Sequences (from 5′ to 3′)	Length	Gene Bank ID
*TNFα*	F: CCC TCA CAC TCA GAT CAT CTT CT	61	NM 013693
	R: GCT ACG ACG TGG GCT ACA G		
*IL-6*	F: AGT CCT TCC TAC CCC AAT TTC C	78	NM 031168
	R: GTC TTG GTC CTT AGC CAC TCC		
*TLR4*	F: GTG CCA GTC AGG GTC ATT CA	119	NM 021297
	R: ACT CCC CAG CCC TTT ATG GA		
*Axin2*	F: TGCATCTCTCTCTGGAGGTG	149	NM 015732.4
	R: TATGTCTTTGCACCAGCCAC		
*CD44*	F: AGCGGCAGGTTACATTCAAA	95	NM 001177787.1
	R: CAAGTTTTGGTGGCACACAG		
*Ctgf*	F: AGCCTCAAACTCCAAACACC	181	NM 010217.2
	R: CAACAGGGATTTGACCAC		
*Gli1*	F: ACCACCCTACCTCTGTCTATT	121	NM 010296.2
	R: TTCAGACCATTGCCCATCAC		
*Hes1*	F: GGTATTTCCCCAACACGGT	101	NM 008235.2
	R: GGCAGACATTCTGGAAATGA		
*Ptprq*	F: CGGAGGTTACTGGAACCGTG	111	NM 008981.3
	R: CAGGGTCCCCACATAGCCT		
*18s*	F: AAG TCC CTG CCC TTT GTA CAC A	70	NR 003278.3
	R: GAT CCG AGG GCC TCA CTA AAC		

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
