# Peer review of "Cytokine-Induced JAK2-STAT3 Activates Tissue Regeneration under Systemic or Local Inflammation"

_ijms, 2022, doi:10.3390/ijms23042262_

Round 1

Reviewer 1 Report

Hello,

The manuscript titled "Cytokine-induced JAK-STAT activates tissue regeneration under systemic or local inflammation" explores the systemic and local inflammation connection and involvement if JAK/STAT pathway on downstream in mouse model. The manuscript presented with well designed control and relevant methods to justify the hypothesis. so good work to authors. 

There is no further comments require for the manuscript.

Thank you,

Author Response

Responses to the referee’s comments

Reviewer #1

The manuscript titled "Cytokine-induced JAK-STAT activates tissue regeneration under systemic or local inflammation" explores the systemic and local inflammation connection and involvement if JAK/STAT pathway on downstream in mouse model. The manuscript presented with well designed control and relevant methods to justify the hypothesis. so good work to authors.

There is no further comments require for the manuscript.

We appreciate your comments.

Reviewer 2 Report

In this paper Kim et.al. have investigated tissue regeneration post inflammation induced via Jak2/STAT3 pathway. One the first read it seems like a nicely done study but on a closer look there are some major issues that need to be addressed. They are as follows:

a) The title/abstract and overall paper needs to be more specific. They are focusing on Jak2/Stat3 so that should be mentioned everywhere. 

b) The phospho-western blots are unreliable as the blocking is not correct. Figure 3 & 4 needs to be repeated. This may change the conclusions and even the narrative of paper and I'm okay with it. 

c) I don't think the Student's t-test is ideal for these experiments. One way-ANOVA would be a better fit. The description of statistical test should be mentioned in the figure legend and also needs to be described in the narrative. Presently, the narrative does not discuss the results in lieu of the statistical test. What is the use of the test if it is not discussed for each result. 

d) Ki67 protein is a classic marker for cells that are dividing found in both cancer and normal cells. Mere cell division cannot be construed as tissue regeneration. There are many other marker that need to be used. This is an over-simplification of a rather very complex process. 

e) Since beta-actin is not the best control for heart samples as evident also in the results presented, I would recommend using something like beta-tubulin for all samples for loading control. As you would be repeating all the western blots so it would be good if you could use a new control. (just a suggestion)

 f) Inflammation as a precursor to tissue regeneration is not new and there are ample marker cells and signaling that are ear-marked exclusively for that process. This paper does not show any markers that are exclusive for tissue regeneration. The experiments convincingly show cell proliferation but to conclude that as tissue regeneration is stretching too far. 

Again, this is a nice study and I look forward to review the revised manuscript. 

Author Response

Responses to the referee’s comments

Reviewer #2

In this paper Kim et.al. have investigated tissue regeneration post inflammation induced via Jak2/STAT3 pathway. One the first read it seems like a nicely done study but on a closer look there are some major issues that need to be addressed. They are as follows:

a) The title/abstract and overall paper needs to be more specific. They are focusing on Jak2/Stat3 so that should be mentioned everywhere.

As suggested, we mentioned JAK2/STAT3 in entire manuscript.

b) The phospho-western blots are unreliable as the blocking is not correct. Figure 3 & 4 needs to be repeated. This may change the conclusions and even the narrative of paper and I'm okay with it.

Skim milk or BSA are usually used for membrane blocking to reduce unspecific antibody binding. However, casein in skim milk cross-react with phospho-protein antibody. Thus, BSA blocking is much better for phospho-protein detection1, 2. For immunoblot in Figure 3A and Figure 4D, we used BSA as a blocking solution as mentioned in Materials and Methods. We have strong confidence that all the blot images are reliable. We appreciate your comment.

c) I don't think the Student's t-test is ideal for these experiments. One way-ANOVA would be a better fit. The description of statistical test should be mentioned in the figure legend and also needs to be described in the narrative. Presently, the narrative does not discuss the results in lieu of the statistical test. What is the use of the test if it is not discussed for each result.

The Student’s t-test is defined as “A statistical test that compares the means and standard deviations of two related or independent samples”. It is widely used for comparisons of two group in immunoblot3 , ELISA4, and IHC5. In this study, statistical analysis was performed using GraphPad Prism 7 software and the differences between control and treated group were analyzed using a Student’s t-test. We re-write the Statistical analysis in detail.

d) Ki67 protein is a classic marker for cells that are dividing found in both cancer and normal cells. Mere cell division cannot be construed as tissue regeneration. There are many other marker that need to be used. This is an over-simplification of a rather very complex process.

We fully agree with your comment. Usually, tissue regeneration is mediated by adult stem cell5-8. To display the tissue regeneration, it is the best to find the self-renewing stem cells in damaged tissue. However, as the adult stem cell marker is still controversial (Lgr5, Bmi1, TERT, Axin2, etc), adult stem cell detection has limitation. In this study, we detected the Ki67+ proliferating cells. Some of Ki67+ cells might be adult stem cells and the rest of Ki67+ cells might be parenchymal cells or others. We compared the number of Ki67+ cells in control vs. treated group and found the increased number. It shows proliferating status clearly. We are sure that the increased Ki67+ cell supports the regenerating capacity.

Moreover, we analyzed five major development pathway target gene expression which is connected with tissue regeneration5 (data not included in manuscript). Sonic hedgehog target gene; Gli1 was mostly upregulated in heart. Wnt/β-catenin target gene; CD44 was highly upregulated in lung and liver. Notch target gene; Hes1 was strongly upregulated in kidney. Hippo target gene; Ctgf was strongly upregulated in duodenum. BMP target gene; Ptprg was upregulated in skin. Those gene upregulation provide the strong evidence of tissue regeneration after inflammatory damage.

Figure 1. qRT-PCR and heatmap. Axin2, CD44, Ctgf, Gli1, Hes, Ptprq mRNA expressions were analyzed in the heart, lung, liver, kidney, duodenum, and skin of animals with chronic or acute systemic and local inflammation. The red color indicates gene upregulation and the green color indicates gene downregulation.

e) Since beta-actin is not the best control for heart samples as evident also in the results presented, I would recommend using something like beta-tubulin for all samples for loading control. As you would be repeating all the western blots so it would be good if you could use a new control. (just a suggestion)

Previous study reported that protein level of β-actin and β-tubulin was various in monkey heart, while protein level of GAPDH was various in mouse heart under ischemic damage condition9. This finding suggests that endogenous control protein should be chosen carefully. As suggested, we compared the endogenous protein level using β-actin, β-tubulin, and GAPDH. We found that strong GAPDH expression and similar level of β-tubulin and β-actin. Thus, β-actin, the endogenous control used in this study are reliable.

Figure 2. Immunoblot. JAK2 and STAT3 phosphorylation as well as endogenous controls; GAPDH, β-tubulin, and β-actin were detected in the heart.

f) Inflammation as a precursor to tissue regeneration is not new and there are ample marker cells and signaling that are ear-marked exclusively for that process. This paper does not show any markers that are exclusive for tissue regeneration. The experiments convincingly show cell proliferation but to conclude that as tissue regeneration is stretching too far.

We analyzed five major development pathway target gene expression which is connected with tissue regeneration5 (data not included in manuscript). Sonic hedgehog target gene; Gli1 was mostly upregulated in heart. Wnt/β-catenin target gene; CD44 was highly upregulated in lung and liver. Notch target gene; Hes1 was strongly upregulated in kidney. Hippo target gene; Ctgf was strongly upregulated in duodenum. BMP target gene; Ptprg was upregulated in skin. Those gene upregulation provide the strong evidence of tissue regeneration after inflammatory damage.

Figure 1. qRT-PCR and heatmap. Axin2, CD44, Ctgf, Gli1, Hes, Ptprq mRNA expressions were analyzed in the heart, lung, liver, kidney, duodenum, and skin of animals with chronic or acute systemic and local inflammation. The red color indicates gene upregulation and the green color indicates gene downregulation.

Again, this is a nice study and I look forward to review the revised manuscript.

Again, we appreciate your comments.

References

  1. Bass JJ, Wilkinson DJ, Rankin D, et al. An overview of technical considerations for Western blotting applications to physiological research. Scand J Med Sci Sports. Jan 2017;27(1):4-25. doi:10.1111/sms.12702
  2. Mahmood T, Yang PC. Western blot: technique, theory, and trouble shooting. N Am J Med Sci. Sep 2012;4(9):429-34. doi:10.4103/1947-2714.100998
  3. Jung YS, Jun S, Kim MJ, et al. TMEM9 promotes intestinal tumorigenesis through vacuolar-ATPase-activated Wnt/beta-catenin signalling. Nat Cell Biol. Dec 2018;20(12):1421-1433. doi:10.1038/s41556-018-0219-8
  4. Tan CW, Chia WN, Qin X, et al. A SARS-CoV-2 surrogate virus neutralization test based on antibody-mediated blockage of ACE2-spike protein-protein interaction. Nat Biotechnol. Sep 2020;38(9):1073-1078. doi:10.1038/s41587-020-0631-z
  5. Suh HN, Kim MJ, Jung YS, Lien EM, Jun S, Park JI. Quiescence Exit of Tert(+) Stem Cells by Wnt/beta-Catenin Is Indispensable for Intestinal Regeneration. Cell Rep. Nov 28 2017;21(9):2571-2584. doi:10.1016/j.celrep.2017.10.118
  6. Post Y, Clevers H. Defining Adult Stem Cell Function at Its Simplest: The Ability to Replace Lost Cells through Mitosis. Cell Stem Cell. Aug 1 2019;25(2):174-183. doi:10.1016/j.stem.2019.07.002
  7. Beumer J, Clevers H. Cell fate specification and differentiation in the adult mammalian intestine. Nat Rev Mol Cell Biol. Jan 2021;22(1):39-53. doi:10.1038/s41580-020-0278-0
  8. Li L, Clevers H. Coexistence of quiescent and active adult stem cells in mammals. Science. Jan 29 2010;327(5965):542-5. doi:10.1126/science.1180794
  9. Nie X, Li C, Hu S, Xue F, Kang YJ, Zhang W. An appropriate loading control for western blot analysis in animal models of myocardial ischemic infarction. Biochem Biophys Rep. Dec 2017;12:108-113. doi:10.1016/j.bbrep.2017.09.001

Reviewer 3 Report

The manuscript is quite well written. The introduction is consistent and informative.  All performed investigations are well described, the results are presented in satisfactory manner. 

Some grammar and spelling mistake have to be corrected.

I would like the Authors to explain why did they perform investigations on groups of just five animals. In fact, currently such experiments are carried out on groups of 8 or 12. 

I am not quite sure if the manuscript meets the scope of the ijms - the Authors did not evaluate the impact of any compounds on the course and aftermath of the inflammation. 

Author Response

Responses to the referee’s comments

Reviewer #3

The manuscript is quite well written. The introduction is consistent and informative.  All performed investigations are well described, the results are presented in satisfactory manner.

Some grammar and spelling mistake have to be corrected.

[1] I would like the Authors to explain why did they perform investigations on groups of just five animals. In fact, currently such experiments are carried out on groups of 8 or 12.

We appreciate your comment. Although we did not inclued the data of serum chemistry and hematology, we performed it to determine the inflammation at day 1 (before treatment) and day 8 (after treatment). To analyzed the serum chemistry and hematology, non-recoverable 1.5 ml blood volume (1.0 + 0.5 ml) is required. We applied ten mice at the beginning and four mice were sacrificed for blood sampling at day 1. Eventually, six mice were remained at day 8 and tissues were collected for immunoblot, gene analysis, and immunohistochemistry. We modified the number of mice in manuscript.

# of sacrificed mice

Ctl

C.LPS

A.LPS

Ova

Hematology

Day 1

2

2

2

2

Day 8

3

3

3

3

Serum chemistry

Day 1

2

2

2

2

Day 8

3

3

3

3

Total

10

10

10

10

[2] I am not quite sure if the manuscript meets the scope of the ijms - the Authors did not evaluate the impact of any compounds on the course and aftermath of the inflammation.

We fully agree with your comment. Based on our previous study, chronic or acute LPS-induced systemic inflammation seems to be attenuated gradually after first induction1. Also, two weeks skin sensitization of ovalbumin (Ova) was found maintaining mild prolonged local and systemic inflammation at day15 2. Although we did not evaluate the impact of LPS or Ova on the course and aftermath of the inflammation, we strongly assume that activated cytokine and intracellular signaling molecules try to overcome the inflammatory damage after LPS or Ova induction in our experimental scope.

References

  1. Suh HN, Kim YK, Lee JY, et al. Dissect the immunity using cytokine profiling and NF-kB target gene analysis in systemic inflammatory minipig model. PLoS One 2021; 16: e0252947. 2021/06/05. DOI: 10.1371/journal.pone.0252947.
  2. Kim YK, Lee JY, Hwang JH, et al. A Pilot Study To Establish an Ovalbumin-induced Atopic Dermatitis Minipig Model. J Vet Res 2021; 65: 307-313. 2021/12/18. DOI: 10.2478/jvetres-2021-0045.

Round 2

Reviewer 2 Report

Majority of the questions were satisfactorily answered by Kim et.al. and I'm satisfied with most of the changes. Yet, the manuscript can be improved further.

Criticism (b) was not relevant on my part as I misread the sentence in the material methods.  

c) All of your results have three groups hence Student’s t-test is not the best option as you can compare only two groups. If you are not comparing within all the three groups then it does not do justice to your analysis. 

d) Very satisfied with the answer and it all makes sense. This information  needs to be a part of the manuscript as it explains a critical part of the manuscript.

e) The authors missed the point on β-actin. The bands are very faded for the heart sample in Fig 4 for the data presented and hence, suggested an alternative control. Should that be considered as a shortcoming in sample processing?

f) This is a satisfactory answer but the present results and the explanation provided in the manuscript does not adequately explain tissue regeneration. Please provide the data that was not shown and add your response to the manuscript to justify your conclusions. 

Without addition of suggested changes in (d & f) that was mentioned in their response, the manuscript in the present state does not explain adequately the tissue regeneration.

I look forward to the revised manuscript. 

Author Response

Responses to the referee’s comments (Round 2)

Reviewer #2

Majority of the questions were satisfactorily answered by Kim et.al. and I'm satisfied with most of the changes. Yet, the manuscript can be improved further.

Criticism (b) was not relevant on my part as I misread the sentence in the material methods. 

We appreciate your comment.

c) All of your results have three groups hence Student’s t-test is not the best option as you can compare only two groups. If you are not comparing within all the three groups then it does not do justice to your analysis.

We fully agree with your comment. If we need to compared three groups together, One way-ANOVA should be used. But, we compared the Ctl vs. C.LPS, Ctl vs. A.LPS, or Ctl vs. Ova individually using Student’s t test as mentioned in figure legend (*P < 0.05 relative to the control). All the statistical analysis in Figures 2A, 3A, 4A, and 4B analyzed same way. Our experimental purpose was to compare the protein expression level, cytokine level, or immune cell number of Ctl vs. treated group. Thus, Student’s t test is reliable.

d) Very satisfied with the answer and it all makes sense. This information needs to be a part of the manuscript as it explains a critical part of the manuscript.

As suggested, we add it in Discussion.

e) The authors missed the point on β-actin. The bands are very faded for the heart sample in Fig 4 for the data presented and hence, suggested an alternative control. Should that be considered as a shortcoming in sample processing?

For some reason, we got fade and week β-actin in heart tissue. However, as there was no expression variation among groups, we decide to use it. We are not 100% sure about the sample processing issue, but lots of cytoskeletons in heart tissue might affect it.

f) This is a satisfactory answer but the present results and the explanation provided in the manuscript does not adequately explain tissue regeneration. Please provide the data that was not shown and add your response to the manuscript to justify your conclusions.

As suggested, we add this data in Figure 4G and mentioned in Results and Discussion.

Without addition of suggested changes in (d & f) that was mentioned in their response, the manuscript in the present state does not explain adequately the tissue regeneration.

Dear Reviewer

We appreciate all your comments. We add the results about the regeneration and it helps a lot to understand our study for other researchers. Again, Thanks.

I look forward to the revised manuscript.

Round 3

Reviewer 2 Report

The second revised version of the manuscript makes an excellent read. The authors added all the extra information asked for and it now makes sense. Congratulations! The manuscript is ready to be published in the present state.